# Global Solver and Its Efficient Approximation for Variational Bayesian Low-rank Subspace Clustering

**Shinichi Nakajima**
Nikon Corporation
Tokyo, 140-8601 Japan
`nakajima.s@nikon.co.jp`

**Akiko Takeda**
The University of Tokyo
Tokyo, 113-8685 Japan
`takeda@mist.i.u-tokyo.ac.jp`

**S. Derin Babacan**
Google Inc.
Mountain View, CA 94043 USA
`dbabacan@gmail.com`

**Masashi Sugiyama**
Tokyo Institute of Technology
Tokyo 152-8552, Japan
`sugi@cs.titech.ac.jp`

**Ichiro Takeuchi**
Nagoya Institute of Technology
Aichi, 466-8555, Japan
`takeuchi.ichiro@nitech.ac.jp`

## Abstract

When a probabilistic model and its prior are given, Bayesian learning offers inference with automatic parameter tuning. However, Bayesian learning is often obstructed by computational difficulty: the rigorous Bayesian learning is intractable in many models, and its variational Bayesian (VB) approximation is prone to suffer from local minima. In this paper, we overcome this difficulty for low-rank subspace clustering (LRSC) by providing an exact global solver and its efficient approximation. LRSC extracts a low-dimensional structure of data by embedding samples into the union of low-dimensional subspaces, and its variational Bayesian variant has shown good performance. We first prove a key property that the VB-LRSC model is highly redundant. Thanks to this property, the optimization problem of VB-LRSC can be separated into small subproblems, each of which has only a small number of unknown variables. Our exact global solver relies on another key property that the stationary condition of each subproblem consists of a set of polynomial equations, which is solvable with the homotopy method. For further computational efficiency, we also propose an efficient approximate variant, of which the stationary condition can be written as a polynomial equation with a single variable. Experimental results show the usefulness of our approach.

## 1 Introduction

Principal component analysis (PCA) is a widely-used classical technique for dimensionality reduction. This amounts to *globally* embedding the data points into a low-dimensional subspace. As more flexible models, the sparse subspace clustering (SSC) [7, 20] and the low-rank subspace clustering (LRSC) [8, 13, 15, 14] were proposed. By inducing sparsity and low-rankness, respectively, SSC and LRSC *locally* embed the data into the union of subspaces. This paper discusses a probabilistic model for LRSC.

As the classical PCA requires users to pre-determine the dimensionality of the subspace, LRSC requires manual parameter tuning for adjusting the low-rankness of the solution. On the other hand,

Bayesian formulations enable us to estimate all unknown parameters without manual parameter tuning [5, 4, 17]. However, in many problems, the rigorous application of Bayesian inference is computationally intractable. To overcome this difficulty, the variational Bayesian (VB) approximation was proposed [1]. A Bayesian formulation and its variational inference have been proposed for LRSC [2]. There, to avoid computing the inverse of a prohibitively large matrix, the posterior is approximated with the matrix-variate Gaussian (MVG) [11].

Typically, the VB solution is computed by the *iterated conditional modes* (ICM) algorithm [3, 5], which is derived through the standard procedure for the VB approximation. Since the objective function for the VB approximation is generally non-convex, the ICM algorithm is prone to suffer from local minima. So far, the global solution for the VB approximation is not attainable except PCA (or the fully-observed matrix factorization), for which the global VB solution has been analytically obtained [17]. This paper makes LRSC another exception with proposed global VB solvers.

Two common factors make the global VB solution attainable in PCA and LRSC: first, a large portion of the degrees of freedom that the VB approximation learns are irrelevant, and the optimization problem can be decomposed into subproblems, each of which has only a small number of unknown variables; second, the stationary condition of each subproblem is written as a polynomial system (a set of polynomial equations).

Based on these facts, we propose an exact global VB solver (EGVBS) and an approximate global VB solver (AGVBS). EGVBS finds all stationary points by solving the polynomial system with the *homotopy method* [12, 10], and outputs the one giving the lowest free energy. Although EGVBS solves subproblems with much less complexity than the original VB problem, it is still not efficient enough for handling large-scale data. For further computational efficiency, we propose AGVBS, of which the stationary condition is written as a polynomial equation with a single variable. Our experiments on artificial and benchmark datasets show that AGVBS provides a more accurate solution than the MVG approximation [2] with much less computation time.

## 2 Background

In this section, we introduce the low-rank subspace clustering and its variational Bayesian formulation.

### 2.1 Subspace Clustering Methods

Let $Y \in \mathbb{R}^{L \times M} = (\boldsymbol{y}_1, \ldots, \boldsymbol{y}_M)$ be $L$-dimensional observed samples of size $M$. We generally denote a column vector of a matrix by a bold-faced small letter. We assume that each $\boldsymbol{y}_m$ is approximately expressed as a linear combination of $M'$ *words* in a dictionary, $D = (\boldsymbol{d}_1, \ldots, \boldsymbol{d}_{M'})$, i.e.,

$$Y = DX + \mathcal{E},$$

where $X \in \mathbb{R}^{M' \times M}$ is unknown coefficients, and $\mathcal{E} \in \mathbb{R}^{L \times M}$ is noise. In subspace clustering, the observed matrix $Y$ itself is often used as a dictionary $D$. The convex formulation of the sparse subspace clustering (SSC) [7, 20] is given by

$$\min_X \|Y - YX\|_{\mathrm{Fro}}^2 + \lambda \|X\|_1, \text{ s.t. } \mathrm{diag}(X) = \boldsymbol{0}, \tag{1}$$

where $X \in \mathbb{R}^{M \times M}$ is a parameter to be estimated, $\lambda > 0$ is a regularization coefficient to be manually tuned. $\| \cdot \|_{\mathrm{Fro}}$ and $\| \cdot \|_1$ are the Frobenius norm and the (element-wise) $\ell_1$-norm of a matrix, respectively. The first term in Eq.(1) requires that each data point $\boldsymbol{y}_m$ can be expressed as a linear combination of a *small* set of other data points $\{\boldsymbol{d}_{m'}\}$ for $m' \neq m$. This *smallness* of the set is enforced by the second ($\ell_1$-regularization) term, and leads to the low-dimensionality of each obtained subspace. After the minimizer $\widehat{X}$ is obtained, $\mathrm{abs}(\widehat{X}) + \mathrm{abs}(\widehat{X}^\top)$, where $\mathrm{abs}(\cdot)$ takes the absolute value element-wise, is regarded as an affinity matrix, and a spectral clustering algorithm, such as normalized cuts [19], is applied to obtain clusters.

In the low-rank subspace clustering (LRSC) or low-rank representation [8, 13, 15, 14], low-dimensional subspaces are sought by enforcing the low-rankness of $X$:

$$\min_X \|Y - YX\|_{\mathrm{Fro}}^2 + \lambda \|X\|_{\mathrm{tr}}. \tag{2}$$

Thanks to the simplicity, the global solution of Eq.(2) has been analytically obtained [8].

## 2.2 Variational Bayesian Low-rank Subspace Clustering

We formulate the probabilistic model of LRSC, so that the maximum a posteriori (MAP) estimator coincides with the solution of the problem (2) under a certain hyperparameter setting:

$$p(Y|A', B') \propto \exp\left(-\frac{1}{2\sigma^2}\|Y - DB'A'^\top\|_{\text{Fro}}^2\right), \tag{3}$$

$$p(A') \propto \exp\left(-\frac{1}{2}\text{tr}(A'C_A^{-1}A'^\top)\right), \qquad p(B') \propto \exp\left(-\frac{1}{2}\text{tr}(B'C_B^{-1}B'^\top)\right). \tag{4}$$

Here, we factorized $X$ as $X = B'A'^\top$, as in [2], to induce low-rankness through the *model-induced regularization* mechanism [17]. In this formulation, $A' \in \mathbb{R}^{M \times H}$ and $B' \in \mathbb{R}^{M \times H}$ for $H \leq \min(L, M)$ are the parameters to be estimated. We assume that hyperparameters

$$C_A = \text{diag}(c_{a_1}^2, \ldots, c_{a_H}^2), \qquad\qquad C_B = \text{diag}(c_{b_1}^2, \ldots, c_{b_H}^2).$$

are diagonal and positive definite. The dictionary $D$ is treated as a constant, and set to $D = Y$, once $Y$ is observed.[1]

The Bayes posterior is written as

$$p(A', B'|Y) = \frac{p(Y|A', B')p(A')p(B')}{p(Y)}, \tag{5}$$

where $p(Y) = \langle p(Y|A', B') \rangle_{p(A')p(B')}$ is the marginal likelihood. Here, $\langle \cdot \rangle_p$ denotes the expectation over the distribution $p$. Since the Bayes posterior (5) is computationally intractable, we adopt the variational Bayesian (VB) approximation [1, 5].

Let $r(A', B')$, or $r$ for short, be a trial distribution. The following functional with respect to $r$ is called the free energy:

$$F(r) = \left\langle \log \frac{r(A', B')}{p(Y|A', B'), p(A')p(B')} \right\rangle_{r(A', B')} = \left\langle \log \frac{r(A', B')}{p(A', B'|Y)} \right\rangle_{r(A', B')} - \log p(Y). \tag{6}$$

In the last equation of Eq.(6), the first term is the Kullback-Leibler (KL) distance from the trial distribution to the Bayes posterior, and the second term is a constant. Therefore, minimizing the free energy (6) amounts to finding a distribution closest to the Bayes posterior in the sense of the KL distance. In the VB approximation, the free energy (6) is minimized over some restricted function space.

### 2.2.1 Standard VB (SVB) Iteration

The standard procedure for the VB approximation imposes the following constraint on the posterior:

$$r(A', B') = r(A')r(B').$$

By using the variational method, we can show that the VB posterior is Gaussian, and has the following form:

$$r(A') \propto \exp\left(-\frac{\text{tr}\left((A' - \widehat{A}')\Sigma_{A'}^{-1}(A' - \widehat{A}')^\top\right)}{2}\right), \qquad r(B') \propto \exp\left(-\frac{(\breve{b}' - \widehat{\breve{b}}')^\top \breve{\Sigma}_{B'}^{-1}(\breve{b}' - \widehat{\breve{b}}')}{2}\right), \tag{7}$$

where $\breve{b}' = \text{vec}(B') \in \mathbb{R}^{MH}$. The means and the covariances satisfy the following equations:

$$\widehat{A}' = \frac{1}{\sigma^2}Y^\top Y \widehat{B}' \Sigma_{A'}, \qquad \Sigma_{A'} = \sigma^2\left(\langle B'^\top Y^\top Y B' \rangle_{r(B')} + \sigma^2 C_A^{-1}\right)^{-1}, \tag{8}$$

$$\widehat{\breve{b}}' = \frac{\breve{\Sigma}_{B'}}{\sigma^2}\text{vec}\left(Y^\top Y \widehat{A}'\right), \quad \breve{\Sigma}_{B'} = \sigma^2\left((\widehat{A}'^\top \widehat{A}' + M\Sigma_{A'}) \otimes Y^\top Y + \sigma^2(C_B^{-1} \otimes I_M)\right)^{-1}, \tag{9}$$

where $\otimes$ denotes the Kronecker product of matrices, and $I_M$ is the $M$-dimensional identity matrix.

For empirical VB learning, where the hyperparameters are also estimated from observation, the following are obtained from the derivatives of the free energy (6):

$$c_{a_h}^2 = \|\widehat{\boldsymbol{a}}_h'\|^2/M + \sigma_{a_h'}^2, \qquad c_{b_h}^2 = \left(\|\widehat{\boldsymbol{b}}_h'\|^2 + \textstyle\sum_{m=1}^M \sigma_{B_{m,h}'}^2\right)/M, \tag{10}$$

$$\sigma^2 = \frac{\mathrm{tr}\left(Y^\top Y\left(I_M - 2\widehat{B}'\widehat{A}'^\top + \left\langle B'(\widehat{A}'^\top\widehat{A}' + M\Sigma_{A'})B'^\top\right\rangle_{r(B')}\right)\right)}{LM}, \tag{11}$$

where $(\sigma_{a_1'}^2, \ldots, \sigma_{a_H'}^2)$ and $((\sigma_{B_{1,1}'}^2, \ldots, \sigma_{B_{M,1}'}^2), \ldots, (\sigma_{B_{1,H}'}^2, \ldots, \sigma_{B_{M,H}'}^2))$ are the diagonal entries of $\Sigma_{A'}$ and $\breve{\Sigma}_{B'}$, respectively. Eqs.(8)–(11) form an ICM algorithm, which we call the standard VB (SVB) iteration.

### 2.2.2 Matrix-Variate Gaussian Approximate (MVGA) Iteration

Actually, the SVB iteration cannot be applied to a large-scale problem, because Eq.(9) requires the inversion of a huge $MH \times MH$ matrix. This difficulty can be avoided by restricting $r(B')$ to be the matrix-variate Gaussian (MVG) [11], i.e.,

$$r(B') \propto \exp\left(-\tfrac{1}{2}\mathrm{tr}\left(\Theta_{B'}^{-1}(B' - \widehat{B}')\Sigma_{B'}^{-1}(B' - \widehat{B}')^\top\right)\right). \tag{12}$$

Under this additional constraint, a gradient-based computationally tractable algorithm can be derived [2], which we call the MVG approximate (MVGA) iteration.

## 3  Global Variational Bayesian Solvers

In this section, we first show that a large portion of the degrees of freedom in the expression (7) are irrelevant, which significantly reduces the complexity of the optimization problem without the MVG approximation. Then, we propose an exact global VB solver and its approximation.

### 3.1  Irrelevant Degrees of Freedom of VB-LRSC

Consider the following transforms:

$$A = \Omega_Y^{\mathrm{right}\top} A', \qquad B = \Omega_Y^{\mathrm{right}\top} B', \qquad \text{where} \qquad Y = \Omega_Y^{\mathrm{left}} \Gamma_Y \Omega_Y^{\mathrm{right}\top} \tag{13}$$

is the singular value decomposition (SVD) of $Y$. Then, we obtain the following theorem:

**Theorem 1** *The global minimum of the VB free energy* (6) *is achieved with a solution such that* $\widehat{A}, \widehat{B}, \Sigma_A, \breve{\Sigma}_B$ *are diagonal.*

(Sketch of proof) After the transform (13), we can regard the observed matrix as a diagonal matrix, i.e., $Y \to \Gamma_Y$. Since we assume Gaussian priors with no correlation, the solution $\widehat{B}\widehat{A}^\top$ is naturally expected to be diagonal. To prove this intuition, we apply a similar approach to [17], where the diagonalities of the VB posterior covariances were proved in fully-observed matrix factorization by investigating perturbations around any solution. We first show that $\widehat{A}'^\top\widehat{A}' + M\Sigma_{A'}$ is diagonal, with which Eq.(9) implies the diagonality of $\breve{\Sigma}_B$. Other diagonalities can be shown similarly. $\square$

Theorem 1 does not only reduce the complexity of the optimization problem greatly, but also makes the problem separable, as shown in the following.

### 3.2  Exact Global VB Solver (EGVBS)

Thanks to Theorem 1, the free energy minimization problem can be decomposed as follows:

**Lemma 1** *Let* $J(\leq \min(L, M))$ *be the rank of* $Y$, $\gamma_m$ *be the* $m$-*th largest singular value of* $Y$, *and*

$$(\widehat{a}_1, \ldots, \widehat{a}_H), (\sigma_{a_1}^2, \ldots, \sigma_{a_H}^2), (\widehat{b}_1, \ldots, \widehat{b}_H), ((\sigma_{B_{1,1}}^2, \ldots, \sigma_{B_{M,1}}^2), \ldots, (\sigma_{B_{1,H}}^2, \ldots, \sigma_{B_{M,H}}^2))$$

*be the diagonal entries of* $\widehat{A}, \Sigma_A, \widehat{B}, \breve{\Sigma}_B$, *respectively. Then, the free energy* (6) *is written as*

$$F = \tfrac{1}{2}\left(LM\log(2\pi\sigma^2) + \tfrac{\sum_{h=1}^J \gamma_h^2}{\sigma^2} + \textstyle\sum_{h=1}^H 2F_h\right), \qquad where \tag{14}$$

---

**Algorithm 1** Exact Global VB Solver (EGVBS) for LRSC.

---

1: Calculate the SVD of $Y = \Omega_Y^{\text{left}} \Gamma_Y \Omega_Y^{\text{right}\top}$.
2: **for** $h = 1$ to $H$ **do**
3:     Find all the solutions of the polynomial system (16)–(18) by the homotopy method.
4:     Discard prohibitive solutions with complex numbers or with negative variances.
5:     Select the stationary point giving the smallest $F_h$ (defined by Eq.(15)).
6:     The global solution for $h$ is the selected stationary point if it satisfies $F_h < 0$, otherwise the null solution (19).
7: **end for**
8: Calculate $\widehat{X} = \Omega_Y^{\text{right}} \widehat{B} \widehat{A}^\top \Omega_Y^{\text{right}\top}$
9: Apply spectral clustering with the affinity matrix equal to $\text{abs}(\widehat{X}) + \text{abs}(\widehat{X}^\top)$.

---

$$2F_h = M \log \frac{c_{a_h}^2}{\sigma_{a_h}^2} + \sum_{m=1}^{J} \log \frac{c_{b_h}^2}{\sigma_{B_{m,h}}^2} - (M+J) + \frac{\widehat{a}_h^2 + M\sigma_{a_h}^2}{c_{a_h}^2} + \frac{\widehat{b}_h^2 + \sum_{m=1}^{J} \sigma_{B_{m,h}}^2}{c_{b_h}^2}$$

$$+ \frac{1}{\sigma^2} \left\{ \gamma_h^2 \left( -2\widehat{a}_h \widehat{b}_h + \widehat{b}_h^2 (\widehat{a}_h^2 + M\sigma_{a_h}^2) \right) + \sum_{m=1}^{J} \gamma_m^2 \sigma_{B_{m,h}}^2 (\widehat{a}_h^2 + M\sigma_{a_h}^2) \right\}, \quad (15)$$

*and its stationary condition is given as follows: for each $h = 1, \ldots, H$,*

$$\widehat{a}_h = \frac{\gamma_h^2}{\sigma^2} \widehat{b}_h \sigma_{a_h}^2, \qquad \sigma_{a_h}^2 = \sigma^2 \left( \gamma_h^2 \widehat{b}_h^2 + \sum_{m=1}^{J} \gamma_m^2 \sigma_{B_{m,h}}^2 + \frac{\sigma^2}{c_{a_h}^2} \right)^{-1}, \quad (16)$$

$$\widehat{b}_h = \frac{\gamma_h^2}{\sigma^2} \widehat{a}_h \sigma_{B_{h,h}}^2, \qquad \sigma_{B_{m,h}}^2 = \begin{cases} \sigma^2 \left( \gamma_m^2 \left( \widehat{a}_h^2 + M\sigma_{a_h}^2 \right) + \frac{\sigma^2}{c_{b_h}^2} \right)^{-1} & (m \leq J), \\ c_{b_h}^2 & (m > J), \end{cases} \quad (17)$$

$$c_{a_h}^2 = \widehat{a}_h^2/M + \sigma_{a_h}^2, \qquad c_{b_h}^2 = \left( \widehat{b}_h^2 + \sum_{m=1}^{J} \sigma_{B_{m,h}}^2 \right) / J. \quad (18)$$

*If no stationary point gives $F_h < 0$, the global solution is given by*

$$\widehat{a}_h = \widehat{b}_h = 0, \qquad \sigma_{a_h}^2, \sigma_{B_{m,h}}^2, c_{a_h}^2, c_{b_h}^2 \to 0 \quad for \ \ m = 1, \ldots, M. \quad (19)$$

Taking account of the trivial relations $c_{b_h}^2 = \sigma_{B_{m,h}}^2$ for $m > J$, Eqs.(16)–(18) for each $h$ can be seen as a polynomial system with $5 + J$ unknown variables, i.e., $\left( \widehat{a}_h, \sigma_{a_h}^2, c_{a_h}^2, \widehat{b}_h, \{\sigma_{B_{m,h}}^2\}_{m=1}^{J}, c_{b_h}^2 \right)$. Thus, Lemma 1 has decomposed the original problem (8)–(10) with $O(M^2 H^2)$ unknown variables into $H$ subproblems with $O(J)$ variables each.

Given the noise variance $\sigma^2$, our exact global VB solver (EGVBS) finds all stationary points that satisfy the polynomial system (16)–(18) by using the *homotopy method* [12, 10],[2] After that, it discards the prohibitive solutions with complex numbers or with negative variances, and then selects the one giving the smallest $F_h$, defined by Eq.(15). The global solution is the selected stationary point if it satisfies $F_h < 0$, or the null solution (19) otherwise. Algorithm 1 summarizes the procedure of EGVBS. If $\sigma^2$ is unknown, we conduct a naive 1-D search by iteratively applying EGVBS, as for VB matrix factorization [17].

### 3.3 Approximate Global VB Solver (AGVBS)

Although Lemma 1 significantly reduced the complexity of the optimization problem, EGVBS is not applicable to large-scale data, since the homotopy method is not guaranteed to find all the solutions in polynomial time in $J$, when the polynomial system involves $O(J)$ unknown variables. For large-scale data, we propose a scalable approximation by introducing an additional constraint that $\gamma_m^2 \sigma_{B_{m,h}}^2$ are constant over $m = 1, \ldots, J$, i.e.,

$$\gamma_m^2 \sigma_{B_{m,h}}^2 = \overline{\sigma}_{b_h}^2 \quad \text{for all} \quad m \leq J. \quad (20)$$

Under this constraint, we obtain the following theorem (the proof is omitted):

**Theorem 2** *Under the constraint* (20)*, any stationary point of the free energy* (15) *for each $h$ satisfies the following polynomial equation with a single variable $\widehat{\widehat{\gamma}}_h$:*

$$\xi_6 \widehat{\widehat{\gamma}}_h^6 + \xi_5 \widehat{\widehat{\gamma}}_h^5 + \xi_4 \widehat{\widehat{\gamma}}_h^4 + \xi_3 \widehat{\widehat{\gamma}}_h^3 + \xi_2 \widehat{\widehat{\gamma}}_h^2 + \xi_1 \widehat{\widehat{\gamma}}_h + \xi_0 = 0, \tag{21}$$

*where*

$$\xi_6 = \frac{\phi_h^2}{\gamma_h^2}, \quad \xi_5 = -2\frac{\phi_h^2 M\sigma^2}{\gamma_h^3} + \frac{2\phi_h}{\gamma_h}, \quad \xi_4 = \frac{\phi_h^2 M^2 \sigma^4}{\gamma_h^4} - \frac{2\phi_h(2M-J)\sigma^2}{\gamma_h^2} + 1 + \frac{\phi_h^2(M\sigma^2 - \gamma_h^2)}{\gamma_h^2}, \tag{22}$$

$$\xi_3 = \frac{2\phi_h M(M-J)\sigma^4}{\gamma_h^3} - \frac{2(M-J)\sigma^2}{\gamma_h} + \frac{\phi_h((M+J)\sigma^2 - \gamma_h^2)}{\gamma_h} - \frac{\phi_h^2 M\sigma^2(M\sigma^2 - \gamma_h^2)}{\gamma_h^3} + \frac{\phi_h(M\sigma^2 - \gamma_h^2)}{\gamma_h}, \tag{23}$$

$$\xi_2 = \frac{(M-J)^2\sigma^4}{\gamma_h^2} - \frac{\phi_h M\sigma^2((M+J)\sigma^2 - \gamma_h^2)}{\gamma_h^2} + ((M+J)\sigma^2 - \gamma_h^2) - \frac{\phi_h(M-J)\sigma^2(M\sigma^2 - \gamma_h^2)}{\gamma_h^2}, \tag{24}$$

$$\xi_1 = -\frac{(M-J)\sigma^2((M+J)\sigma^2 - \gamma_h^2)}{\gamma_h} + \frac{\phi_h MJ\sigma^4}{\gamma_h}, \quad \xi_0 = MJ\sigma^4. \tag{25}$$

*Here, $\underline{\gamma} = (\sum_{m=1}^J \gamma_m^{-2}/J)^{-1}$ and $\phi_h = \left(1 - \frac{\gamma_h^2}{\underline{\gamma}^2}\right)$. For each real solution $\widehat{\widehat{\gamma}}_h$ such that*

$$\widehat{\gamma}_h = \widehat{\widehat{\gamma}} + \gamma_h - \frac{M\sigma^2}{\gamma_h}, \quad \widehat{\kappa} = \gamma_h^2 - (M+J)\sigma^2 - \left(M\sigma^2 - \gamma_h^2\right)\phi_h \frac{\widehat{\widehat{\gamma}}}{\gamma_h}, \tag{26}$$

$$\widehat{\tau} = \frac{1}{2MJ}\left(\widehat{\kappa} + \sqrt{\widehat{\kappa}^2 - 4MJ\sigma^4\left(1 + \phi_h\frac{\widehat{\widehat{\gamma}}}{\gamma_h}\right)}\right), \quad \widehat{\delta}_h = \frac{\sigma^2}{\sqrt{\widehat{\tau}}}\left(\gamma_h - \frac{M\sigma^2}{\gamma_h} - \widehat{\gamma}_h\right)^{-1}, \tag{27}$$

*are real and positive, the corresponding stationary point candidate is given by*

$$\left(\widehat{a}_h, \sigma_{a_h}^2, c_{a_h}^2, \widehat{b}_h, \overline{\sigma}_{b_h}^2, c_{b_h}^2\right) = \left(\sqrt{\widehat{\gamma}\widehat{\delta}}, \frac{\sigma^2\widehat{\delta}_h}{\gamma_h}, \sqrt{\widehat{\tau}}, \sqrt{\widehat{\gamma}/\widehat{\delta}}/\gamma_h, \frac{\sigma^2}{\gamma_h\widehat{\delta}_h - \phi_h\frac{\sigma^2}{\sqrt{\widehat{\tau}}}}, \sqrt{\widehat{\tau}}/\gamma_h^2\right). \tag{28}$$

Given the noise variance $\sigma^2$, obtaining the coefficients (22)–(25) is straightforward. Our approximate global VB solver (AGVBS) solves the sixth-order polynomial equation (21), e.g., by the 'roots' function in MATLAB®, and obtain all candidate stationary points by using Eqs.(26)–(28). Then, it selects the one giving the smallest $F_h$, and the global solution is the selected stationary point if it satisfies $F_h < 0$, otherwise the null solution (19). Note that, although a solution of Eq.(21) is not necessarily a stationary point, selection based on the free energy discards all non-stationary points and local maxima. As in EGVBS, a naive 1-D search is conducted for estimating $\sigma^2$.

In Section 4, we show that AGVBS is practically a good alternative to the MVGA iteration in terms of accuracy and computation time.

## 4  Experiments

In this section, we experimentally evaluate the proposed EGVBS and AGVBS. We assume that the hyperparameters $(C_A, C_B)$ and the noise variance $\sigma^2$ are unknown and estimated from observations. We use the full-rank model (i.e., $H = \min(L, M)$), and expect VB-LRSC to automatically find the true rank without any parameter tuning.

We first conducted an experiment with a small artificial dataset ('artificial small'), on which the exact algorithms, i.e., the SVB iteration (Section 2.2.1) and EGVBS (Section 3.2), are computationally tractable. Through this experiment, we can measure the accuracy of the efficient approximate variants, i.e., the MVGA iteration (Section 2.2.2) and AGVBS (Section 3.3). We randomly created $M = 75$ samples in $L = 10$ dimensional space. We assumed $K = 2$ clusters: $M^{(1)*} = 50$ samples lie in a $H^{(1)*} = 3$ dimensional subspace, and the other $M^{(2)*} = 25$ samples lie in a $H^{(2)*} = 1$ dimensional subspace. For each cluster $k$, we independently drew $M^{(k)*}$ samples from $\mathcal{N}_{H^{(k)*}}(\mathbf{0}, 10 I_{H^{(k)*}})$, where $\mathcal{N}_d(\boldsymbol{\mu}, \Sigma)$ denotes the $d$-dimensional Gaussian, and projected them into the observed $L$-dimensional space by $R^{(k)} \in \mathbb{R}^{L \times H^{(k)*}}$, each entry of which follows $\mathcal{N}_1(0, 1)$. Thus, we obtained a noiseless matrix $Y^{(k)*} \in \mathbb{R}^{L \times M^{(k)*}}$ for the $k$-th cluster. Concatenating all clusters, $Y^* = (Y^{(1)*}, \ldots, Y^{(K)*})$, and adding random noise subject to $\mathcal{N}_1(0, 1)$ to each entry gave an artificial observed matrix $Y \in \mathbb{R}^{L \times M}$, where $M = \sum_{k=1}^K M^{(k)*} = 75$. The *true* rank of $Y^*$

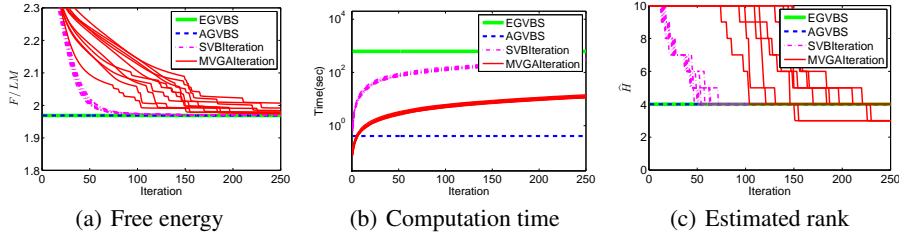

(a) Free energy &emsp; (b) Computation time &emsp; (c) Estimated rank

Figure 1: Results on the 'artificial small' dataset ($L = 10, M = 75, H^* = 4$). The clustering errors were $1.3\%$ for EGVBS, AGVBS, and the SVB iteration, and $2.4\%$ for the MVGA iteration.

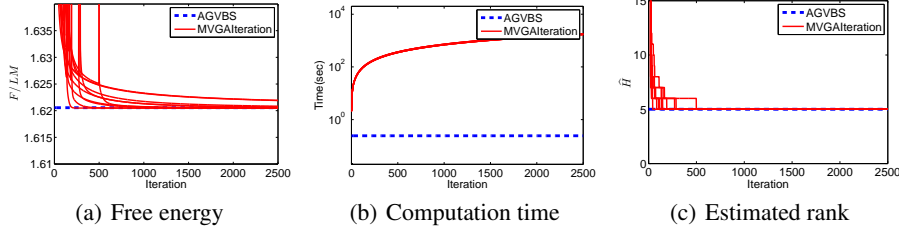

(a) Free energy &emsp; (b) Computation time &emsp; (c) Estimated rank

Figure 2: Results on the 'artificial large' dataset ($L = 50, M = 225, H^* = 5$). The clustering errors were $4.0\%$ for AGVBS and $11.2\%$ for the MVGA iteration.

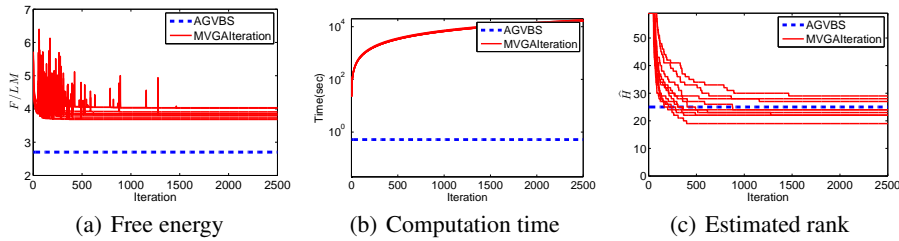

(a) Free energy &emsp; (b) Computation time &emsp; (c) Estimated rank

Figure 3: Results on the '1R2RC' sequence ($L = 59, M = 459$) of the Hopkins 155 motion database. The clustering errors are shown in Figure 4.

is given by $H^* = \min(\sum_{k=1}^{K} H^{(k)*}, L, M) = 4$. Note that $H^*$ is different from the rank $J$ of the observed matrix $Y$, which is almost surely equal to $\min(L, M) = 10$ under the Gaussian noise.

Figure 1 shows the free energy, the computation time, and the estimated rank of $\widehat{X} = \widehat{B}'\widehat{A}'^\top$ over iterations. For the iterative methods, we show the results of 10 trials starting from different random initializations. We can see that AGVBS gives almost the same free energy as the exact methods (EGVBS and the SVB iteration). The exact method requires a large computation cost: EGVBS took 621 sec to obtain the global solution, and the SVB iteration took $\sim 100$ sec to achieve almost the same free energy. The approximate methods are much faster: AGVBS took less than 1 sec, and the MVGA iteration took $\sim 10$ sec. Since the MVGA iteration had not converged after 250 iterations, we continued the MVGA iteration until 2500 iterations, and found that the MVGA iteration sometimes converges to a local solution with significantly higher free energy than the other methods. EGVBS, AGVBS, and the SVB iteration successfully found the *true* rank $H^* = 4$, while the MVGA iteration sometimes failed. This difference is actually reflected to the clustering error, i.e., the misclassification rate with all possible cluster correspondences taken into account, after spectral clustering [19] is performed: $1.3\%$ for EGVBS, AGVBS, and the SVB iteration, and $2.4\%$ for the MVGA iteration.

Next we conducted the same experiment with a larger artificial dataset ('artificial large') ($L = 50, K = 4, (M^{(1)*}, \dots, M^{(K)*}) = (100, 50, 50, 25), (H^{(1)*}, \dots, H^{(K)*}) = (2, 1, 1, 1)$), on which EGVBS and the SVB iteration are computationally intractable. Figure 2 shows results with AGVBS and the MVGA iteration. An advantage in computation time is clear: AGVBS took $\sim 0.1$ sec, while the MVGA iteration took more than 100 sec. The clustering errors were $4.0\%$ for AGVBS and $11.2\%$ for the MVGA iteration.

Finally, we applied AGVBS and the MVGA iteration to the *Hopkins 155 motion* database [21]. In this dataset, each sample corresponds to a trajectory of a point in a video, and clusteirng the trajectories amounts to finding a set of rigid bodies. Figure 3 shows the results on the '1R2RC'

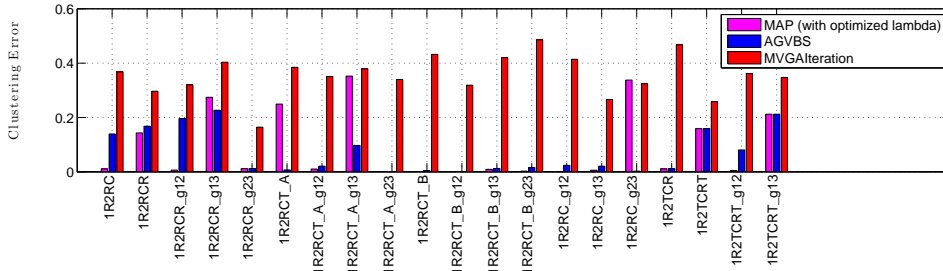

Figure 4: Clustering errors on the first 20 sequences of Hopkins 155 dataset.

($L = 59, M = 459$) sequence.[3] We see that AGVBS gave a lower free energy with much less computation time than the MVGA iteration. Figure 4 shows the clustering errors over the first 20 sequences. We find that AGVBS generally outperforms the MVGA iteration. Figure 4 also shows the results with MAP estimation (2) with the tuning parameter $\lambda$ optimized over the 20 sequences (we performed MAP with different values for $\lambda$, and selected the one giving the lowest average clustering error). We see that AGVBS performs comparably to MAP with optimized $\lambda$, which implies that VB estimates the hyperparameters and the noise variance reasonably well.

## 5   Conclusion

In this paper, we proposed a global variational Bayesian (VB) solver for low-rank subspace clustering (LRSC), and its approximate variant. The key property that enabled us to obtain a global solver is that we can significantly reduce the degrees of freedom of the VB-LRSC model, and the optimization problem is separable.

Our exact global VB solver (EGVBS) provides the global solution of a non-convex minimization problem by using the homotopy method, which solves the stationary condition written as a polynomial system. On the other hand, our approximate global VB solver (AGVBS) finds the roots of a polynomial equation with a single unknown variable, and provides the global solution of an approximate problem. We experimentally showed advantages of AGVBS over the previous scalable method, called the matrix-variate Gaussian approximate (MVGA) iteration, in terms of accuracy and computational efficiency. In AGVBS, SVD dominates the computation time. Accordingly, applying additional tricks, e.g., parallel computation and approximation based on random projection, to the SVD calculation would be a vital option for further computational efficiency.

LRSC can be equipped with an outlier term, which enhances robustness [7, 8, 2]. With the outlier term, much better clustering error on Hopkins 155 dataset was reported [2]. Our future work is to extend our approach to such robust variants. Theorem 2 enables us to construct the *mean update* (MU) algorithm [16], which finds the global solution with respect to a large number of unknown variables in each step. We expect that the MU algorithm tends to converge to a better solution than the standard VB iteration, as in robust PCA and its extensions. EGVBS and AGVBS cannot be applied to the applications where $Y$ has missing entries. Also in such cases, Theorem 2 might be used to derive a better algorithm, as the VB global solution of fully-observed matrix factorization (MF) was used as a subroutine for partially-observed MF [18].

In many probabilistic models, the Bayesian learning is often intractable, and its VB approximation has to rely on a local search algorithm. Exceptions are the fully-observed MF, for which an analytic-form of the global solution has been derived [17], and LRSC, for which this paper provided global VB solvers. As in EGVBS, the homotopy method can solve a stationary condition if it can be written as a polynomial system. We expect that such a tool would extend the attainability of global solutions of non-convex problems, with which machine learners often face.

### Acknowledgments

The authors thank the reviewers for helpful comments. SN, MS, and IT thank the support from MEXT Kakenhi 23120004, the FIRST program, and MEXT KAKENHI 23700165, respectively.

## Footnotes

[1] Our formulation is slightly different from the one proposed in [2], where a clean version of $Y$ is introduced as an additional parameter to cope with outliers. Since we focus on the LRSC model without outliers in this paper, we simplified the model. In our formulation, the clean dictionary corresponds to $YBA^\top(BA^\top)^\dagger$, where $\dagger$ denotes the pseudo-inverse of a matrix.

[2] The homotopy method is a reliable and efficient numerical method to solve a polynomial system [6, 9]. It provides all the isolated solutions to a system of $n$ polynomials $\boldsymbol{f}(\boldsymbol{x}) \equiv (f_1(\boldsymbol{x}), \ldots, f_n(\boldsymbol{x})) = \boldsymbol{0}$ by defining a smooth set of homotopy systems with a parameter $t \in [0, 1]$: $\boldsymbol{g}(\boldsymbol{x}, t) \equiv (g_1(\boldsymbol{x}, t), g_2(\boldsymbol{x}, t), \ldots, g_n(\boldsymbol{x}, t)) = \boldsymbol{0}$, such that one can continuously trace the solution path from the easiest ($t = 0$) to the target ($t = 1$). We use HOM4PS-2.0 [12], which is one of the most successful polynomial system solvers.

[3]Peaks in free energy curves are due to pruning, which is necessary for the gradient-based MVGA iteration. The free energy can jump just after pruning, but immediately get lower than the value before pruning.

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
