[Reviews · NeurIPS 2013]

Submitted by Assigned_Reviewer_1

[Summary]
The paper presents an exact algorithm to find globally optimal solution for variational Bayesian Low-rank Subspace Clustering and an efficient algorithm to approximate the solution. The major contribution is to discover that the coefficient and covariance matrices at the optimal point, which consists of prohibitively numerous elements, can be diagonalized using the singular value decomposition of the observation matrix Y, and that the diagonal elements are given by one of the solutions of a system of polynomial equations with the (nearly) same number of variables than the rank of the observation matrix Y. Based on the properties, an algorithm to compute the exact solution is derived. Then, by imposing a simple constraint, the system of polynomial equations is approximately decomposed into a set of independent equations with a single variable, to derive a computationally efficient algorithm. These algorithms were empirically evaluated using artificial data sets and a realistic data set cited from an open-source database. The results show the applicability and efficiency to the large-dimensional data.
[Major Comments]
Even though the main result might be an increment from the references [2] and [18], the paper has originality and quality enough potentially to attract an interest from a subset of the NIPS community who are working on the statistical analysis of big data. As for the clarity, the paper is well organized enough to easily understand the concepts behind the algorithms. The minor concern is whether the algorithms can tolerate the observation dimensionality more than 10 thousands such as gene expression and fMRI brain imaging data, because recently such data are increasing more and more and they are potentially suitable targets of the proposed algorithms. Also, to reinforce the quality of the contribution, I would like to give some minor comments as below.
[Minor Comments]
* Is there any reason why D appears in Eq.(3)? As long as the paper focuses on LRSC after Eq.(2), it seems better that D is replaced by Y to avoid the redundancy.
* Is there any possibility to parallelize the algorithm in order to improve computational time by cluster computers? Such a discussion will helpful to researchers who are interested in applications.

[For author feedback]
Thanks for the feedback. Especially, I understood the reason for the variable D very well (even though it is a little bit strange).
Summary: Except for minor comments, there is no critical issue on either quality, clarity, or originality.

Submitted by Assigned_Reviewer_2

The problem of low rank subspace clustering can be addressed by variational Bayes, with the advantage over MAP methods that the rank is adjusted automatically, and virtually no free parameters have to be tuned. The posterior variances also possibly give more information.

As reviewed by the authors (I am not familiar with this literature), subspace clustering seems to optimize for an affinity matrix to be put into spectral clustering, by essentially regressing datapoints onto themselves with low rank constraint. The method here tackles the first step, how to obtain the affinity matrix. This can be done by variational Bayes. Given that the error is measured by Frobenius norm and that all observations are complete (I wonder how realistic that is), recent results [18] can be adapted to solve this VB problem by a singular value decomp. of the data matrix plus mapping the SVs one by one, using some complicated polynomial equations, which however can be solved analytically. In particular after a 2nd constraint, the latter is just O(L), no iterations are needed. The global optimum is found, even including hyperparameters (prior variances).

Given that (a) subspace clustering is a useful problem, and (b) VB is a good method for doing it,
the results here are very good and very useful. I cannot comment on (a), (b), they cite some
papers in computer vision. I am a big fan of [18]. What is done here is closely related, but some additional new ideas (homotopy method) are needed. Just as with [18], it is easy to understand the basic idea just by looking at the Frobenius error, but very hard to check the details: I assume they are correct.

The experimental evaluation is convincing, but they could make a larger effort describing the
context. What is clustered here? What is the clustering error? In order to save space, throw out the equations on iterative VB: nobody needs them anymore after this paper! Also, please give proof sketches. Here, you do not even state clearly why X diagonalizes in the right SV basis -- this is easy to argue for.

Also, why did you not compare against the simpler MAP solution of (2)? Previous work did not seem to employ VB at all, except for [2] (non-empty overlap with the authors here?), so I find it odd to only compare against a rather esoteric variant -- if another globally solvable method is available.

Finally, the main limitation of all this work is that I cannot have a weighted error, in particular the data matrix Y has to be completely observed. It may be good to comment on whether this is an issue in the application here.
Summary: Extends recent work on global analytical solution of VB matrix factorization (densely observed,
uniform Gaussian noise) to the problem of low-rank subspace clustering. Essentially, the setup is
constrained enough so that one gets the solution by transforming (shrinking) the singular values of the data matrix. Given that this is an important problem, the results are very convincing.

Submitted by Assigned_Reviewer_5

This paper applies variational Bayesian (VB) for low-rank subspace clustering (LRSC). The major contribution is a global solver and an efficient approximation. By carefully analyzing the redundancy in VB-LRSC, the inference problem is decomposed into subproblems, which can all be efficiently solved by homotopy method. Experiments show that the proposed method achieves significantly lower clustering error compared with the state of the art.

I think the decomposition and efficient solution to the subproblems are interesting and useful. The experiment is also quite encouraging. So I recommend accepting this paper.
Summary: The proposed algorithm observes an interesting decomposition in the variational Bayes method for low-rank subspace clustering. Efficient global inference is hence achieved, with encouraging experimental results.
Author Feedback

Author rebuttal: Dear Reviewers,

Thank you very much for your sensible comments on our manuscript.
Please find our answers to the questions below.

With best regards,
Authors


Reply to Reviewer 1:

- Applicability to very big data (with dimensionality more than 10 thousands),
and possibility to parallelize the algorithm.

SVD dominates the computation time of AGVBS,
so the use of parallel SVD is a vital option to handle big data;
also the use of an approximate algorithm, e.g., based on random projection
is another possible option.
We will add this discussion in the final version.

- Why D appears in (3)?

Eq.(3) is a distribution function of Y, and must be normalized so that
the integral over Y is equal to one.
If we would replace D with Y in (3), which becomes a random variable,
we need an additional normalization constant, which depends on A' and B',
and thus the MAP estimator does not coincide with the solution to (2).
Therefore, we assume that D is a constant in (3), and it is replaced with Y later,
to make our probabilistic model consistent with (2).


Reply to Reviewer 2:

- About clustering in the experiments.

In the experiment on artificial data, the observed samples are clustered into groups.
Since the "true" clustering is known, we can calculate the clustering error
by taking all possible cluster correspondences into account.
In the experiment on the Hopkins 155 dataset, each sample is a trajectory
of a point in the video. In this case, clustering the trajectories amounts
to finding the set of rigid bodies.
We will add a few sentences to explain the experimental setup in more detail,
saving the space by omitting the equations for the iterative VB algorithms,
as suggested.

- Sketch of proof.

Thank you for your suggestion.
We will extend the sketch of proof to give more intuition.

- Why did not we compare against the MAP solution?

MAP does not allow us to estimate the noise variance.
Therefore, we need to manually adjust lambda in (2) to get a reasonable result.
The estimated rank can be optimized by scanning lambda,
which leads to a good clustering result.
Since what we argue in this paper is that our VB solver gives a reasonable estimated rank
with no manually tuned parameter,
we did not compare VB to MAP with optimized lambda in the original manuscript.
However, we now think that the best possible accuracy by MAP with the
optimized lambda would be informative, to show how accurate the rank estimated by our VB solver is.
Accordingly, we will add results with MAP to the final version.


- Weighted error and missing entries.

Both EGVBS and AGVBS cannot be applied to the case of weighted error or missing entries.
We will add a sentence that clearly states this fact,
and also discuss possibilities of extending our results to such cases.
Actually, the following paper used the result in [18] to develop an efficient
local search algorithm for the case of incomplete observations:

M. Seeger and G. Bouchard, "Fast Variational Bayesian Inference for Non-Conjugate Matrix Factorization Models," AISTATS2012.

We expect that a similar approach can be applied for subspace clustering.
We will add these comments to the final version.


Reply to Reviewer 5:

Thanks for your positive comments.
We will further polish the paper, following the suggestions by other reviewers.